# Enigmatic Surface Ruptures at Cape Rytyi and Surroundings, Baikal Rift, Siberia: Seismic Hazard Implication

Oksana V. Lunina [1,*] , Ivan A. Denisenko [1], Anton A. Gladkov [1,2] and Carlos Braga [3]

1 Institute of the Earth's Crust, Siberian Branch of Russian Academy of Sciences, Lermontova Street 128, 664033 Irkutsk, Russia

2 Center for the Development of Continuing Education of Children, Ministry of Education of Irkutsk Region, Sergeeva Street 5/6, 664043 Irkutsk, Russia

3 Department of Earth Sciences, Carleton University, 1125 Colonel By Drive Ottawa, Ottawa, ON K1S 5B6, Canada

* Correspondence: lounina@crust.irk.ru; Tel.: +7-9148852409

**Abstract:** The geomorphic expression of active faulting and distinction of paleoseismic events in areas that are rapidly obscured by erosion/sedimentation still remains a considerable scientific problem. The present article discusses the revealing of surface faulting ruptures and their parameters to identify capable faults without trenching and to estimate the magnitude of earthquakes. The case study was at Cape Rytyi, located in Baikal-Lena Nature Reserve on the northwestern shore of Lake Baikal. Based on unmanned aerial photography, GPR, and structural observations, we mapped and investigated the relation between geomorphological forms and ruptures. The obtained results show that past landslides and paleoruptures at Cape Rytyi and its surroundings are associated with at least two earthquakes. The Mw of the earlier event was 7.3 (Ms = 7.4); the Mw of the later one was 7.1 (Ms = 7.3). The paleoruptures in the distal part of the delta of the Rita River and on the southeastern slope of the Baikal Ridge were included in the seismogenic rupture zone, which traces some 37 km along the Kocherikovsky fault. The approximate intervals in which earthquakes occurred are 12–5 ka and 4–0.3 ka, respectively. The applied analysis methods can be useful for paleoseismology and assessing seismic hazards in similar regions elsewhere.

**Keywords:** rupture; delta; UAV; geomorphological mapping; ground-penetrating radar; structural geology; earthquake; Baikal rift





## 1. Introduction

The problem of identifying active faults and their displacement parameters in various geological settings is crucial in the design of large engineering infrastructures [1,2]. The cost of the object and the possibility of its construction depend on the results of a detailed geological mapping and accurate seismic hazard assessment. Seismic hazards are defined as "the potential for dangerous, earthquake-related natural phenomena such as ground shaking, fault rupture, or soil liquefaction" [3] or "a property of an earthquake that can cause damage and loss" [4,5]. In any propedeutic survey, it is of utmost importance to determine whether certain linear morphological and/or structural features are associated with seismic fault ruptures. Their association with an active fault zone could imply the possibility of generating strong linear morphogenic earthquakes [6–9]. The definition of active fault depends on the regulatory rules of different countries and is considered in detail by Carbonel et al. [10], who describe various stratigraphic, geomorphic, structural, and chronologic scenarios and their possibilities and limitations for unambiguously determining whether a fault is active.

Once the evidence for the seismogenic origin of the landscape is verified, the question arises if the scarp formed during one or several seismic events. This determines a more correct estimate of a single-event displacement, which in turn affects the assessment of the

magnitude of the associated earthquake. Trenching provides direct observation of faults and the displacements along them [2,11–14], but its application is not always practicable for various reasons. For example, the deep and large-volume trench excavated in the Ruesta Fault in Spain had a total cost of 57,000 euros [10]. The work required the use of two large backhoes and trucks to extract the excavated sediment. Beyond the costs issue, large excavations are not possible along the coasts of Lake Baikal, where almost the entire territory belongs to national parks and nature reserves. On the other hand, that area is highly seismic. About 2000–3000 earthquakes are recorded annually by 20 stations [15]. The 1862 M ~ 7.5 Thagan shock was the largest historical event, which was well described on Lake Baikal [16]. Several earthquakes with M > 6 occurred in the region after 1950. A complex pattern of 30 km-long surface ruptures formed during the 1957 Ms = 7.6 Muya event northeast of Lake Baikal, and numerous secondary coseismic effects were after the 1959 Ms = 6.8 earthquake in Middle Baikal [17].

In addition, the Baikal area contains many enigmatic structures, the genesis of which is unclear and attracted the attention of both specialists and laypeople. The purpose of the present work is to study the near-surface geology, geomorphology, and sediment deformations at Cape Rytyi on the northwestern coast of Lake Baikal (Figure 1) and, using this example, to show the possibilities of combining drone aerial photography, ground-penetrating radar (GPR), and structural observations to identify coseismic surface ruptures, determine their relative age in relation to other geomorphological features, and reconstruct of displacements with subsequent calculation of the seismic potential of the active fault, without disturbing the soil cover.

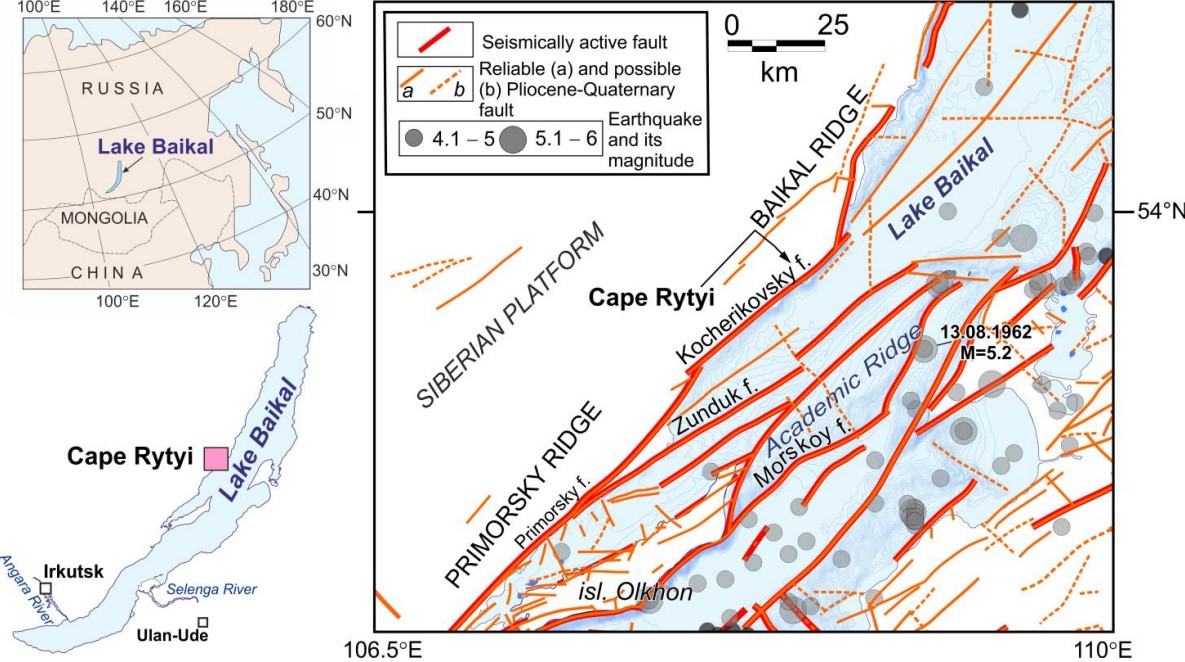

**Figure 1.** Location map of Cape Rytyi (**left**) and its tectonic framework (**right**). Epicenters of instrumentally recorded earthquakes with a magnitude $M \geq 4.1$ during 1950–2019, according to [10], are shown (labeled date is for the event mentioned in text).

## 2. Study Area

Cape Rytyi is the most mysterious place on the northwestern coast of Lake Baikal (Figures 1 and 2). The basin is known as the central section of the Baikal Rift zone, repeatedly described in [18–21]. Geological information about Cape Rytyi can be found in popular science publications, on the geological map, and online [22–24]. Cape Rytyi coastal plain is built of debris flow and deltaic deposits, mainly coarse alluvium of the Rita River, which often changes its course, forming potholes. The length of the cape is ca. 4.32 km in the northeastern direction and ca. 1.68 km in the northwestern direction.

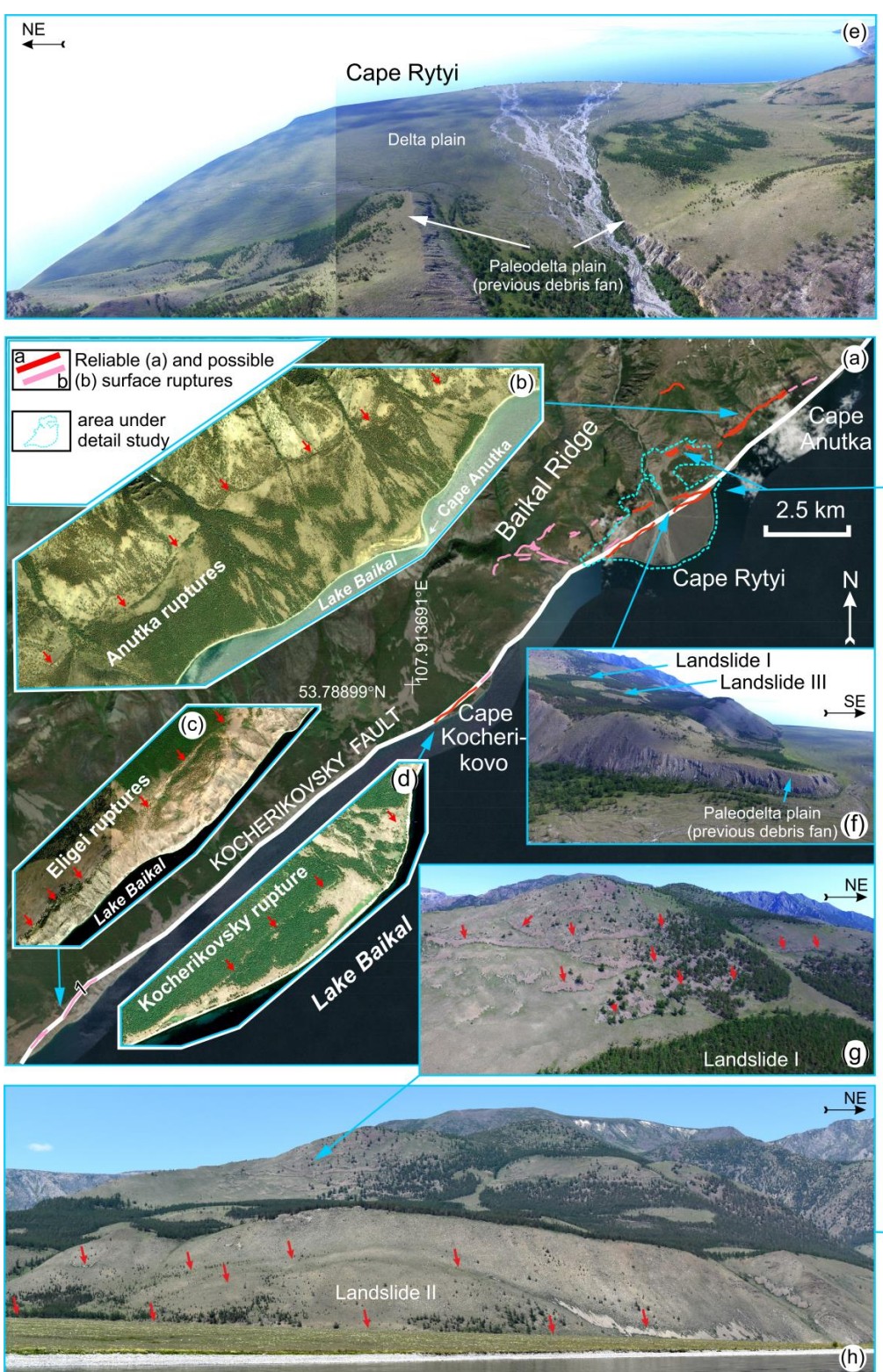

**Figure 2.** Seismogenic surface ruptures derived from interpretation of satellite data along the Kocherivsky fault: (**a**–**d**) on an image from YandexMap; (**e**–**h**) fragments of Cape Rytyi. The red arrows show the paleoruptures.

Cape Rytyi is known for its anomalous magnetic field, elevated residual isostatic gravity anomalies [22], traces of powerful debris flows, and landslide deformations along seismogenic rupture zones (Figure 2h). From time to time, the GPS signal vanishes in the

surroundings of Cape Rytyi, which is reflected in the loss of connection between the unmanned aerial vehicle (UAV) and the control panel, rezeroing the coordinates on the display of a handheld GPS and navigation equipment of the vessel. The yearly freezing of Lake Baikal is regularly accompanied by the formation of a crack parallel to the cape coastline.

From the northwest, the delta Rita is framed by the slopes of the Baikal Ridge, which is composed of Proterozoic metamorphic sandstones, shales, and granites [24], covered in places by slope detritus. At the exit of the river from the mountains, deposits of the previous alluvial cone (Figure 2f), composed of mudflow deposits lying on ancient alluvium, are observed on the sides of the valley. Bedrocks are intensely mylonitized and cataclased along the Kocherikovsky fault, traced along the coast of Lake Baikal from the Glubokaya Pad River valley to Cape Shartlay (Figures 1 and 2). Figures in [25,26] show the main seismogenic fault along the northwestern boundary of the delta plain. Based on the preservation of deformations, in comparison with other dated fault scarps of the Baikal rift, Chipizubov et al. [26] estimated the age of the last faulted offset ranging from six to eight thousand years BP. A series of shorter ruptures are also mapped in the Baikal Ridge on the left of the Rita River (Figure 2g). On the left bank of the Rita River, a seismically-induced landslide with a vertical throw of up to 400 m is described [25]. However, during the geomorphological analysis of the relief in the present work, this deformation was not explicitly recognized. In addition, no other faults were recorded at Cape Rytyi and its vicinity before our studies. Moreover, the main seismogenic fault in the distal part of the Rita River delta is strongly eroded, and in places where it adjoins the slope, it is covered with talus, which complicates its identification as a surface rupture. Thus, to understand the nature of the landforms in the vicinity of Cape Rytyi and to determine the seismic potential of the Kocherikovsky fault using displacements, more detailed studies were needed.

## 3. Materials and Methods

### 3.1. Remote Sensing Methods

Before starting the detailed structural and geomorphological studies of the area of Cape Rytyi, we deciphered seismogenic surface ruptures on satellite images along the entire Kocherikovsky fault (Figure 2). Various images are presented on Google Earth and Yandex. Web map services, as well as images from the Pleiades-1A/1B spacecraft with a resolution of 0.5 m/pixel from 15 June 2016 and 6 November 2016, were used. Consequently, considering the critical analysis of the materials [25,26], reliable and possible seismogenic surface ruptures were identified over a distance of ~37 km. In the southwest, 3.2–6.9 km from the mouth of the Eligei River, the faults separate two blocks of diabase and crystalline schist that underwent seismogravitational subsidence (Figure 2c). Further to the northeast, the disruptions are masked by the steep shore of Lake Baikal, running under the water. The ruptures again reappear after 14.5 km at Cape Kocherikovo and the slopes of the Baikal Ridge (Figure 2d), and then at Cape Rytyi and in the vicinity of Cape Anyutka (Figure 2b). Based on the consistency of the strike of the seismogenic ruptures associated with the Kocherikovsky fault, they likely belong to a single zone of Holocene deformations.

Further research was focused on Cape Ryty, where we conducted unmanned aerial photography and processed photographic material to obtain a detailed orthophoto and digital surface model (DSM). The latter, due to the development of computer power and the possibility of obtaining ultra-detailed images, were increasingly used in the study of surface deformations [27–32]. Aerial photography is carried out using a DJI Phantom 4 Pro V2.0 UAV equipped with a 1-inch CMOS matrix with a resolution of 20 MP and a mechanical shutter for preventing image distortion. In the relatively flat terrain on the cape, the unmanned aerial vehicle was controlled automatically. In territories with complicated terrain characterized by sharp changes in elevation, the control was manual.

The flight altitude is no higher than 120–150 m relative to the Earth's surface at a flight velocity of no more than 30 km/h. The georeferencing of the obtained cartographic materials (DSM and orthophotomaps) is performed using the photography center coordinates recorded by the embedded GPS receiver of the UAV and the ground control points

(markers) distributed along the coastal and central parts of Cape Rytyi. As a result of aerial photography conducted on June 30, July 1, and July 5 of 2019, more than 7000 photographs were processed using structure-from-motion photogrammetry [33] realized in "Agisoft Metashape" licensed software [34]. The obtained results were converted to GeoTIFF and *.jpeg formats.

The obtained orthoimagery of 6–10 cm/pixel (Figure 3, see also: http://activetectonics. ru/content/AFS-Tiles/Ortho/Rytiy2019/leaflet.html accessed on 15 March 2023 in the author's geoportal [35]), DSM of 10–20 cm/pixel (Figure 4), terrestrial and aerial photographs taken with lateral composition, field observations, and numerous topographic profiles, in places where trees and shrubs on DSM were absent, were used to build a detailed geomorphological map. In addition, the DSM was useful for measuring vertical displacements of the original surfaces (OS) across the strike of the seismic fault scarp, which is a standard approach in the study of active faults [11,12,36,37].

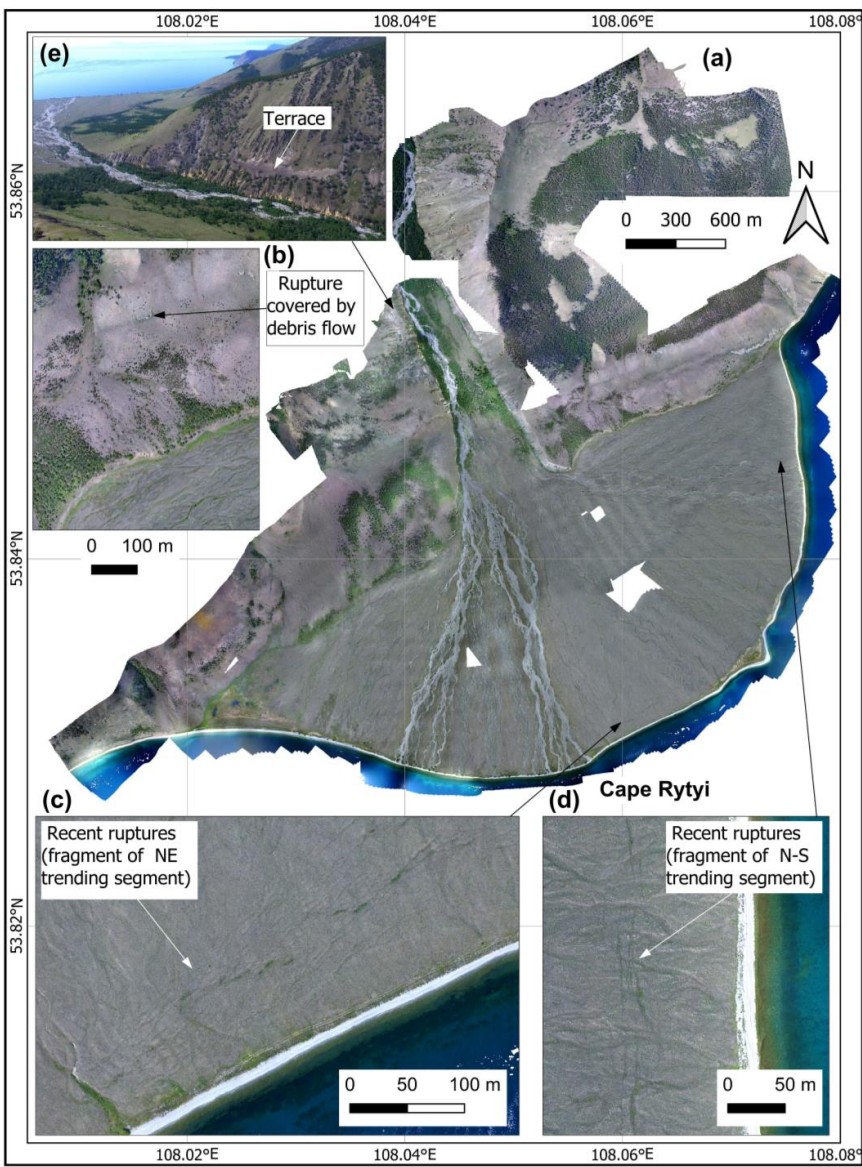

**Figure 3.** (**a**) Orthophotomap of Cape Rytyi; (**b**–**d**) its fragments; (**e**) location of the Rita River terrace of ~39 m in height according to the aerial photography data from the summer of 2019.

The mapping stage with final graphic enhancement was performed using the QGIS 3.22.0 software. The cartographic symbolization system and colors were designed in line with the principles of map graphic design, such as legibility, visual contrast, figure-ground

organization, and hierarchical composition [38,39]. We were relatively free to experiment while choosing the optimal graphic solutions. The proposed map legend is rich in color symbols emphasizing landform genesis and structures because it was important for the determination of the relative time of the fault formation.

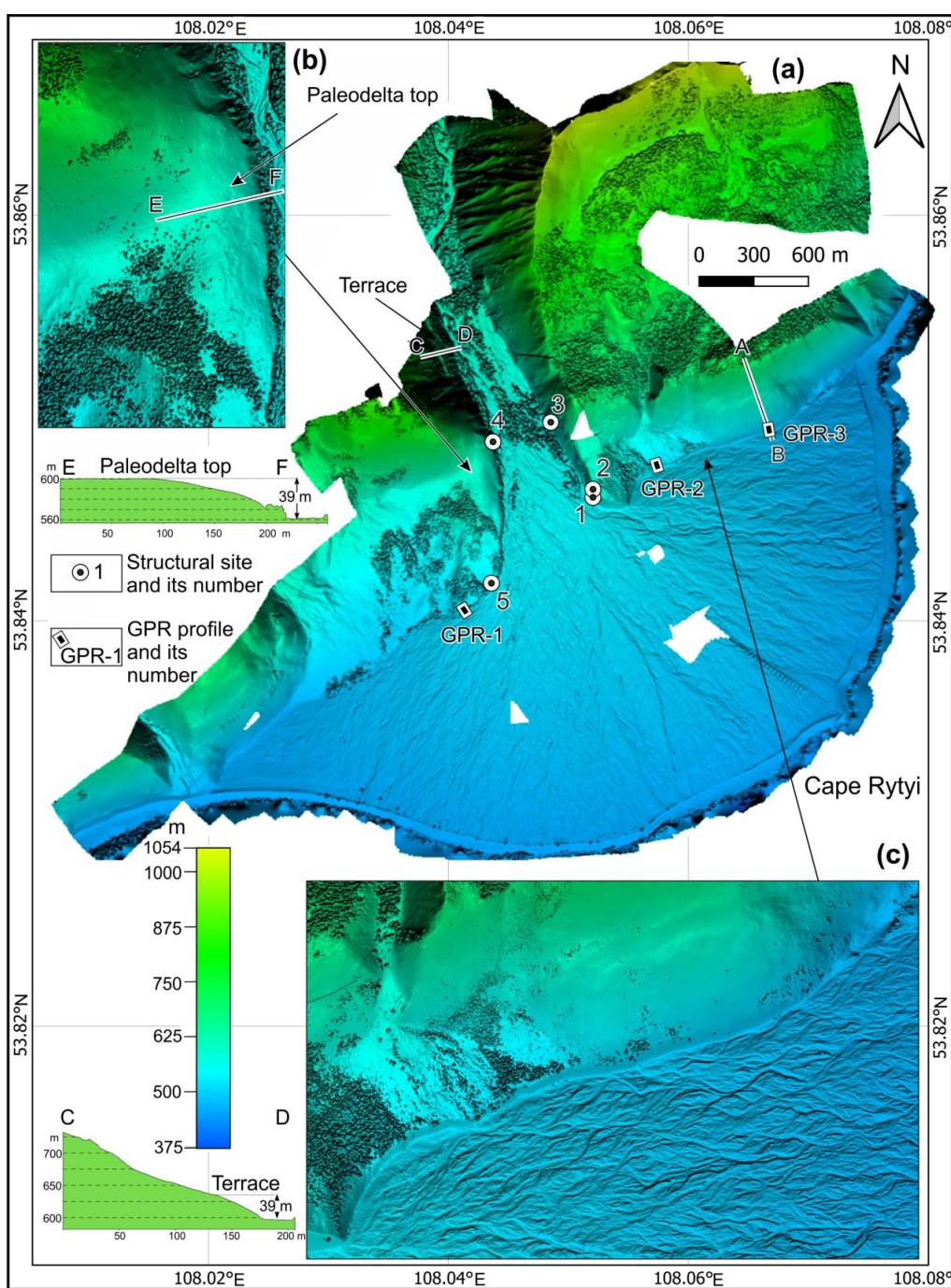

**Figure 4.** (**a**) Location of structural observation sites, GPR lines, and the hypsometric profiles along the A–B (discussed bellow), C–D, and E–F lines on DSM of Cape Rytyi; (**b**,**c**) details of DSM. The terrace is shown in Figure 3e.

### 3.2. Surface Research Method and Observations

Among the terrestrial methods for studying paleoseismic ruptures, we used GPR surveys, as well as geomorphological, geological, and structural observations. The observations in the available outcrops were performed to find evidence of seismically induced deformation in the rocks and sediments. We documented five sites (Figure 4), one of

which consists of Proterozoic granite, and the others are composed of Upper Quaternary sediments. We have paid special attention to the discontinuities in pebbles and boulders of debris-flow deposits and alluvium of the first fan of the Rita River. Azimuth and dip angles of 239 fractures were measured and analyzed on the fracturing diagrams, which were plotted on the upper hemisphere of a Wulff net with a counting circle size of 1% and density contours every 1%.

GPR lines were surveyed across the strike of the mapped paleoruptures in the distal part of the Rita River delta (Figure 4). Among other geophysical methods, the GPR is the most suitable tool for paleoseismic studies [2,12,40]. With the correct selection of antennas, depending on the geotechnical conditions of the terrain, section structure, proposed displacement values, and ground electrophysical parameters, it can measure offsets close to the real displacements [16,41–45]. In addition, the GPR is a noninvasive method that is very important for using it in the Baikal-Lena Nature Reserve.

At three sites chosen as key GPR images from all those obtained and considered in the present work, we used the Logis-Geotech OKO-2 radar and an unshielded ABDL-Triton antenna with a 100 MHz dipole transmitter providing a penetration of 16 m and a resolution of 0.5 m. The ABDL-Triton antenna was designed to work on rough ground traverses. In practice, the penetration in our geotechnical conditions is 1–2 m less than the technical specifications declare.

Considering the relief on GPR profiles, topographic data are gathered along the survey lines using an electronic tacheometer Leica, with map spacing between successive measurements between 0.5 and 1.5 m. The ground elevation points were then input into GeoScan 32 software applied for processing GPR records. Processing began with the choice of signal amplification, brightness, and contrast. Then, if necessary, we calibrated zero to the surface, analyzed local noise, and inputted the elevation. The dielectric permittivity ($\varepsilon$), which is the basic parameter to assess the penetration depth, was determined as 7.5 from analyzing tilted linear noise and hyperboles on radargrams. The next step consisted of standard processing procedures, with a low pass or band pass filtering for the reduction or removal of noise and inverse filtering for improving depth resolution. When interpreting the radargrams, we applied principles of seismic stratigraphy analysis [46] and structural geology.

Finally, we compared vertical displacements, based on GPR data and the morphostructural analysis of the fault scarp, to conclude if the measured surface offsets in DSM are the result of a single or multi-events. Then we used the topographic profiling on DSM, which is the easiest method of documenting the vertical components along the paleoseismic fault and searching maximum and middle offsets [36], and estimated the magnitudes Ms and Mw of the associated paleoarthquake, applied relationships from works [47–49]. We calculated both magnitude types because the surface-wave magnitude scale (Ms) is the most commonly used method of estimating the size of shallow earthquakes [50], and it saturates only around magnitude 8 [51]. Such great events happen about once a year on average for the whole planet and are atypical for extension zone such as the Baikal Rift.

## 4. Results

### 4.1. Geomorphological Mapping

The constructed geomorphological map made it possible to obtain an initial representation of the most prominent morphological features in the vicinity of Cape Rytyi and the occurrence of associated seismic ruptures (Figure 5). Among the morphological features, fluvial and endogenous landforms are the most widespread. The latter are mainly represented by remnants of tectonic slopes, which are abundant in the Baikal Rift [52]. At Cape Rytyi, they are significantly complicated by denudation and gravitational processes.

Paleoseismic scarps with gentle to steep slope angles, ranging from 10° to 78°, are delimited by a coseismic rupture at their base and are presumed as younger tectonic landforms. Based on their relationship with other morphological elements, two generations of Upper Pleistocene–Holocene faults, which were produced during different reactivation phases, were identified. Some surface deformations on the slopes of the Baikal Range are covered

by talus and debris flow deposits or destroyed by erosional processes (Figures 3 and 5, ruptures R-2, R-3, R-5, R-6, R-7, R-8, and R-13). In contrast, other ruptures, also partly subject to denudation and erosion, in the distal part of the Rita River delta, cut across debris flow cones and landslide body II but are buried under recent delta deposits (Figures 3–5, ruptures R-9, R-10, and R-11).

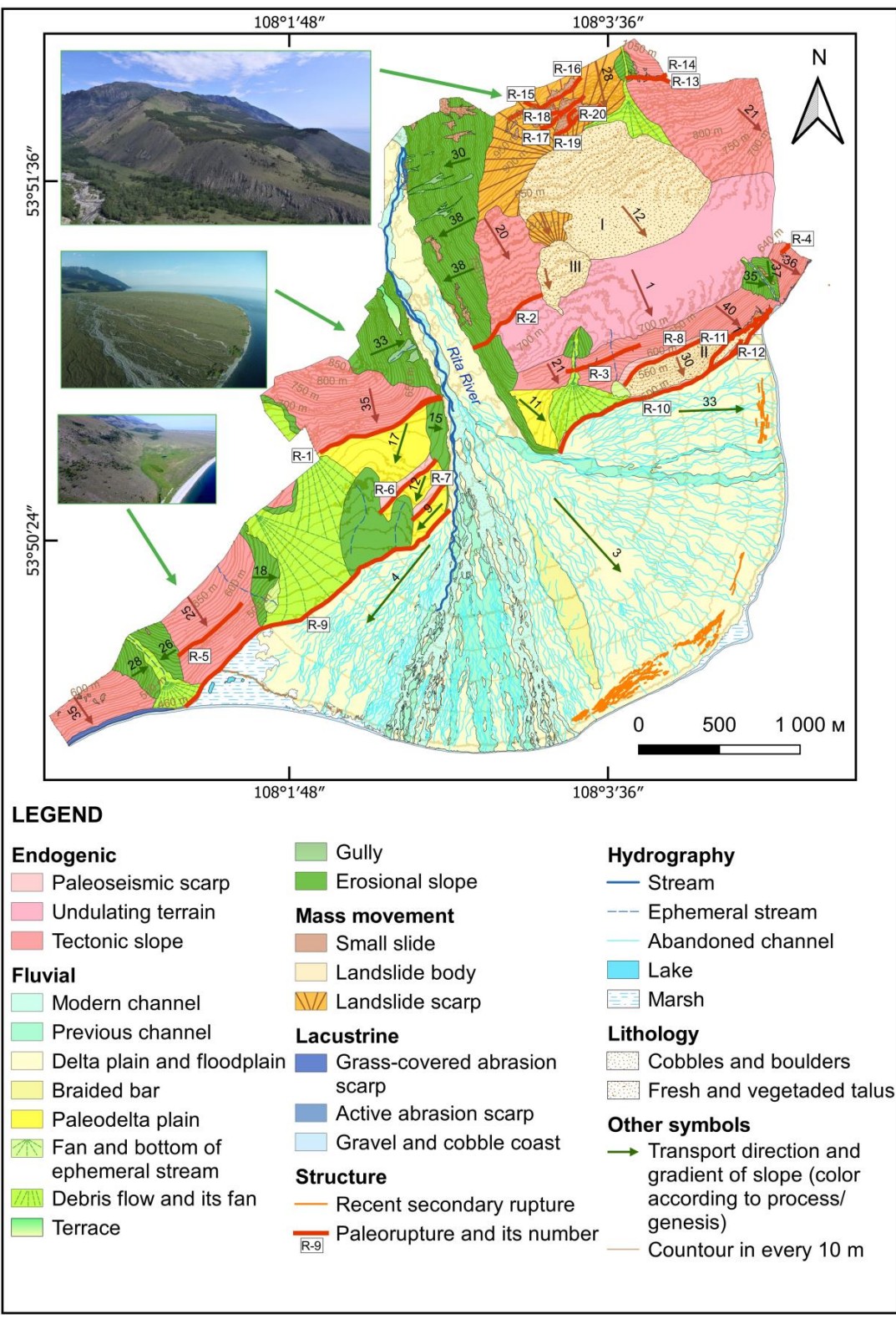

**Figure 5.** Geomorphological map of the Rita delta and adjacent area.

In addition to the described seismically induced ruptures, in the peripheral part of the Rita River delta, over 2.9 km, we mapped very recent fractures, which formed 30–150 m from the shore of the lake as a secondary effect of the M = 5.2 earthquake that occurred on 13 August 1962 (Figure 3c,d, see also: http://activetectonics.ru/content/AFS-Tiles/DEM/Rytiy2019DEM/leaflet.html accessed on 15 March 2023 on the author's geoportal [35]).

The epicenter of the seismic event was located 35 km from Cape Rytyi, in the zone of the Morskoy Fault (Figure 1). Evidence of the connection of the modern fractures with the 1962 seismic event is given in our previous article [53], in which we showed that an earthquake initiated the formation of the fractures. After that, a subsidence of coarse clastic deposits of the Rita River delta in the coastal zone occurred.

Slope failures complementary to the Upper Pleistocene–Holocene ruptures were also discerned in the past earthquakes in the study area. Based on the significant forest coverage and the indistinct manifestation, landslide I (Figure 2f) occurred before the first generation of paleoseismic ruptures, which instead are coeval with landslide II (Figure 2h). During the formation of the second generation of paleoseismic features (ruptures R-9 and R-10), repeated displacements occurred along some early faults, most likely near landslide II and on the right side of the Rita River (Figure 5). The small landslide III could have been formed together with later paleoseismic ruptures since it collapsed after the formation of landslide I and covered the earlier generation rupture R-2 on the left side of the Rita River (Figure 5).

On both sides of the Rita valley, remnants of a previous phase are preserved (Figures 2f, 6 and 7). They mainly consist of debris flows of various generations (Figure 6) overlying alluvial deposits. The latter are exposed on the right bank of the Rita River (Figure 7e–g). Partially, the previous fan is buried under younger debris flows that emerge along neighboring minor valleys. Most parts of the paleodelta were affected by faulting, and the hanging-wall block was thus downthrow and buried under the younger sediments of the modern delta.

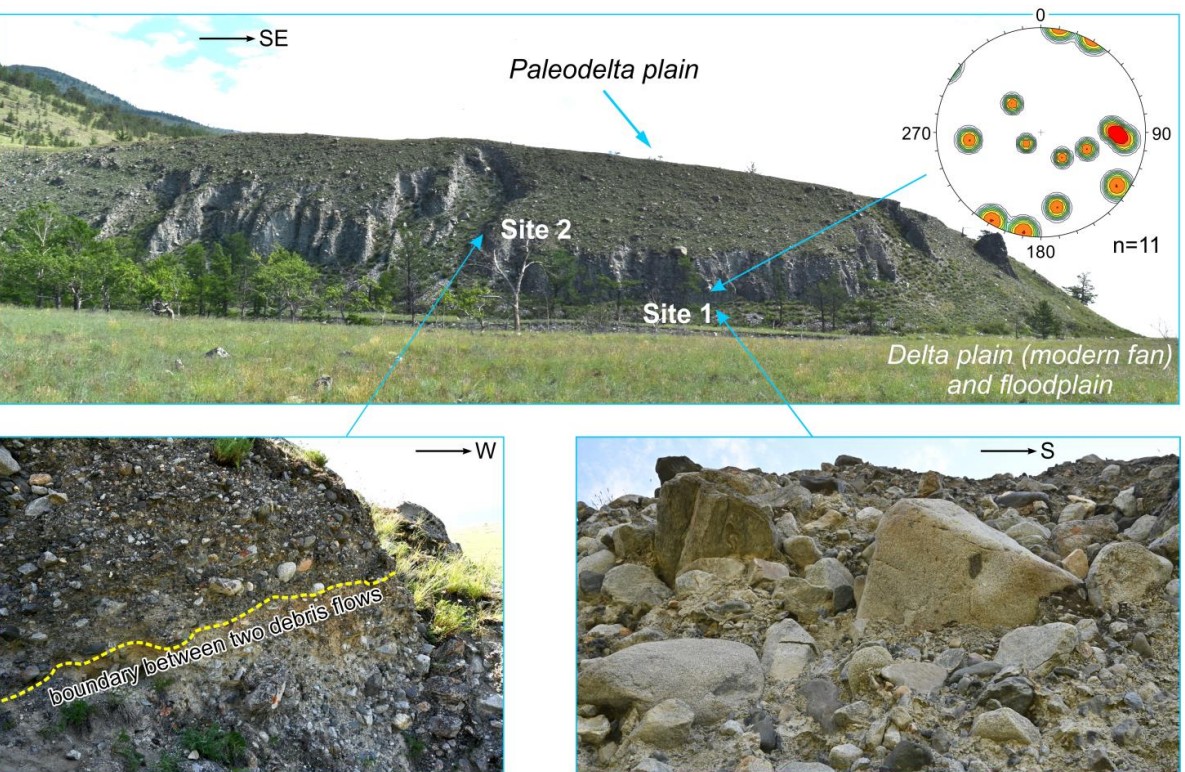

**Figure 6.** Debris flow deposits in sites 1 and 2 (see location in Figure 4) and diagram of the fractures in clasts (equal angle upper hemisphere projection; *n*—number of measurements; counting circle size 1%; intensity isolines are 1, 2, 3 and >; different colors show density concentration from 0 (white) to >9 (red)).

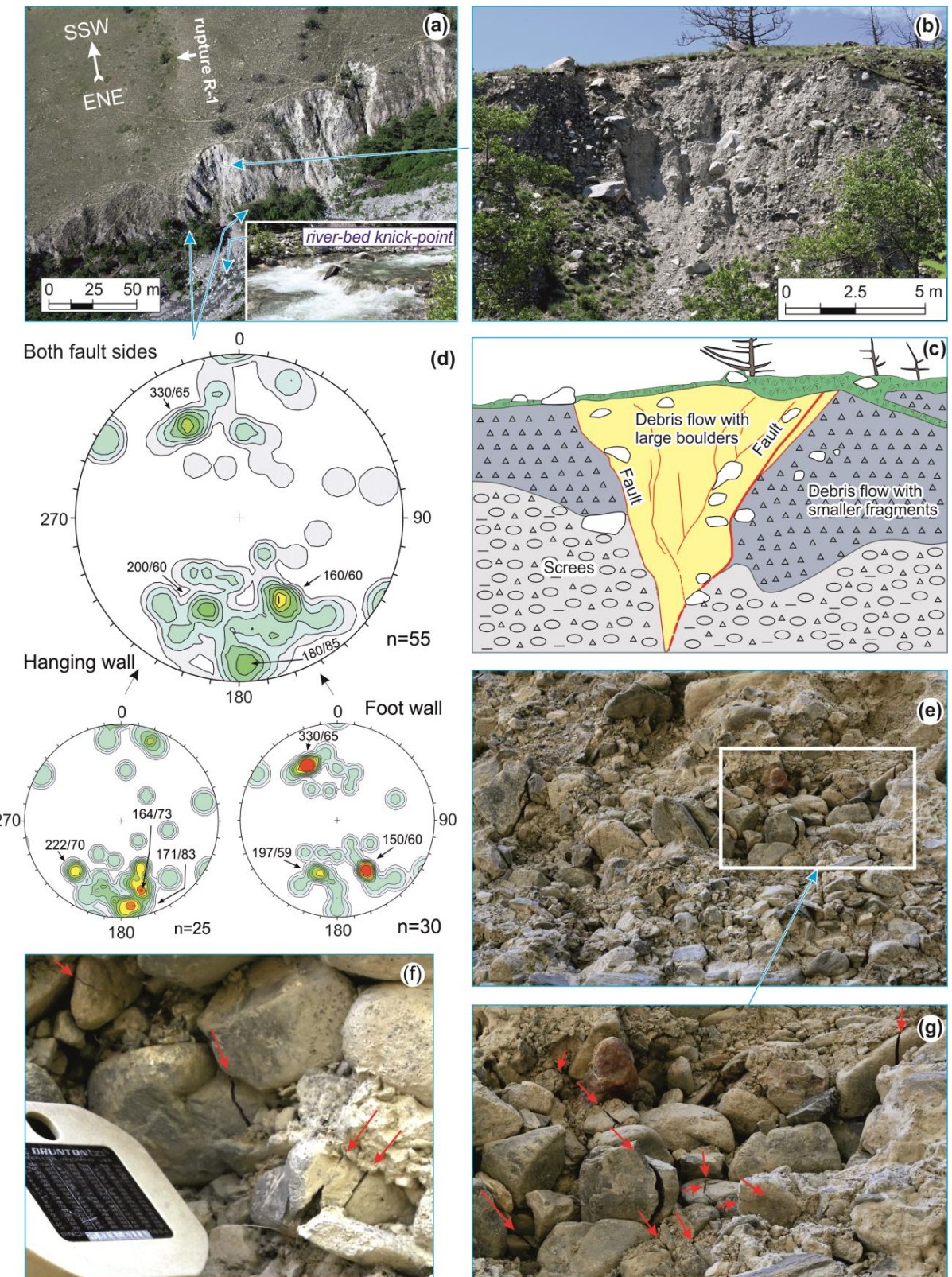

**Figure 7.** Site 4: (**a**) Shear zone R-1; (**b,c**) light brown cemented debris flow deposits filled the rupture (see location in Figure 4); (**d**) diagrams of fractures in the alluvium underlying debris flow deposits (Wulff net upper hemisphere projection; n—number of measurements; counting circle size 1%; intensity isolines are 1, 2, 3 and >; different colors show density concentration from 0 (white) to >9 (red)); (**e–g**) fractures in alluvium shown by red arrows.

### 4.2. Fracturing in Outcrops

The fracturing pattern in the area of Cape Rytyi is heterogeneous. At observation sites 1 (53.84533° N; 108.05258° E) and 2 (53.84582° N; 108.05254° E) located 200–220 m from the nearest paleorupture R-10 (Figures 4 and 5), deposits are characterized by various generations of mudflows, which include both well-rounded alluvium and angular fragments ranging in size from 1 cm to 2 m in diameter (Figure 6). The clay matrix is ~30% of the outcrop. Fractures in the clasts are very rare here and do not form regular fracture systems (see diagram in Figure 6).

Debris flow deposits on the left bank of the Rita River are underlain by fractured Precambrian granite outcropping at site 3 (53.84909° N; 108.04903° E, Figure 8). Moreover, there are no morphological features for its continuation from the right to the left bank. However, the fresh appearance of fracturing and the external loosening of the massif indicate that the rocks have experienced shaking. Furthermore, the top curve of the surface is suggestive of a graben like the ones described in Nevada after the earthquakes of 16 December 1954 [54]. The main fracture systems trend predominantly NE, NNE, and N-S, typical of the Baikal rift zone.

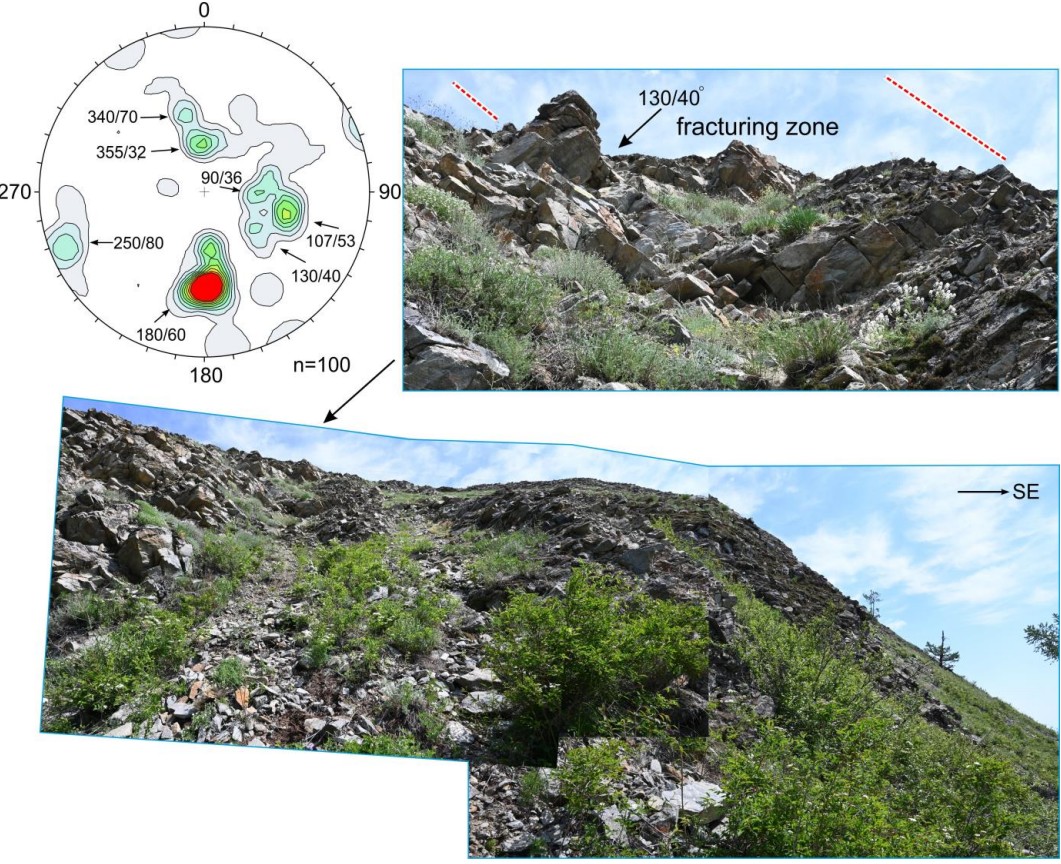

**Figure 8.** Outcrop of granite rocks in site 3 and fractures diagram (see location in Figure 4, Wulff net upper hemisphere projection; n—number of measurements; counting circle size 1%; intensity isolines are 1, 2, 3 and >; different colors show density concentration from 0 (white) to >9 (red)).

Paleorupture R-1, trending towards an average direction of 075° in Quaternary sediments, exposes at site 4 (53.84807° N; 108.04424° E) opposite the granite outcrop (Figures 5 and 7a). Closer to the river valley, it splits into branches and has a nearly E–W direction corresponding to the most intense fracture system in the diagram for granites (Figure 8). The discontinuity is manifested by a gentle scarp with a repose angle of 19°, paired with strips of vegetation, suggesting the occurrence of aligned fractures characterized by wetter soil conditions (Figure 7a). At site 4, the shear zone R-1 is 1–1.5 m wide

in the lower part and up to 7 m in the upper part, and consists of light brown strongly cemented debris flow deposits with angular fragments from 1 cm to 2 m in size and a large amount of clay matrix (Figure 7b). These deposits within the zone are cut by a clear system of long-length fractures (Figure 7b,c), indicating two deformation events affecting this fault. A clastic dike filled the gap that was formed during the first paleoearthquake, while during the second one, large cracks were formed inside this dike. No vertical displacements were observed in this outcrop.

The host sediments in the upper part of the section are also represented by debris flows, but their clasts size is smaller, and their ratio with the matrix is different. In the lower part, the sediments overlie on intensely fractured and loosened alluvium of the Rita River (Figure 7e–g). The main fracture systems affecting the pebbles trend northeastward and nearly E–W (Figure 7d), with the former dominating in the footwall and the latter in the hanging wall. If the maxima in the diagram representing the two fault sides (330°/65° and 160°/60°) are considered conjugate systems, we may assume that they were formed under NW–SE tensile conditions. In this stress field, left-lateral displacements occurred along the E–W trending shears, and normal offsets happened along the NE trending fractures. Already 20 m from the main rupture R1 (Figure 7a–c) downstream of the Rita River, cracks in pebbles were not observed.

Intense fracturing in poorly sorted and cemented pebbles and boulders is also documented at site 5 (53.84138° N; 108.04457° E), located on the right bank of the Rita River in correspondence with rupture R-9) bordering the modern delta (Figures 4, 5 and 9). The scarp has undergone erosional processes, as evidenced by the significant curvature of its upper crest.

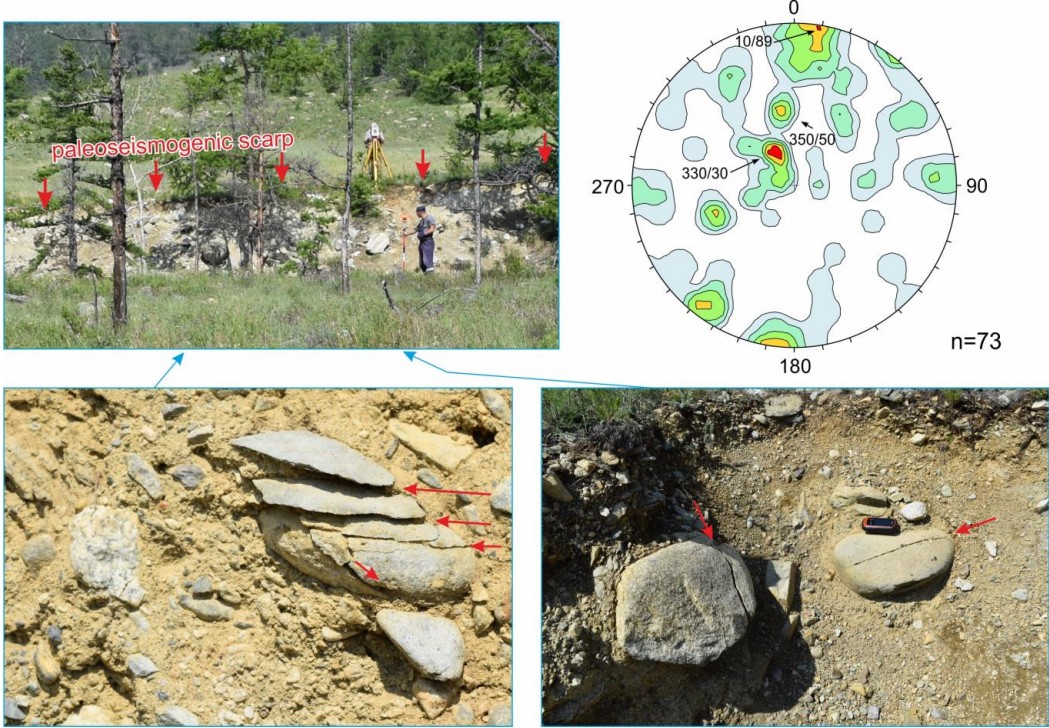

**Figure 9.** Fractures in pebbles shown by red arrows in site 5, which is located on the seismogenic scarp associated with rupture R-9, and fractured diagram (see location in Figure 4, Wulff upper hemisphere projection; n—number of measurements; counting circle size 1%; intensity isolines are 1, 2, 3 and >; >; different colors show density concentration from 0 (white) to >9 (red)).

The sediments in the exposed wall are poorly sorted alluvium and slope debris ranging in size from a few millimeters to 0.7 m, cemented by clay cement (Figure 9). A fine fraction and a reddish tint of sediments dominate in the upper part of the outcrop and a gray coarse-grained fraction in the lower one. A trough up to 4 m in width, which was apparently

used by watercourses, was preserved at the base of the scarp. The main, nearly E–W and NE trending fractures with a dip to the north and northwest, respectively, are recognized within the otherwise chaotic fracture system.

In general, fracture analysis of the observed Quaternary sediments indicates that the fracture intensity, as measured in pebbles and boulders, strongly depends on their proximity to the paleoseismic rupture. The fractures become rare only at a 10 m distance from the main rupture, and at 100 m, isolated fractured pebbles do not generate any clear trend (Figure 6). The predominant orientation of the fractures in the pebbles and the intensity of their appearance confirm the trend of paleoseismic ruptures mapped by remote sensing methods.

*4.3. GPR Research*

4.3.1. GPR Profile 1

The 25 m-long GPR profile begins at coordinates 53.83976° N, 106.687° E, it cuts across the scarp on the right bank of the Rita River, 265 m from site 5 to the west-southwest, and ends at coordinates 53.83968° N, 108.04211° E (Figure 10a). Three radar facies were encountered; the first is characterized by a chaotic wave pattern and corresponds to reddish, weakly cemented, and loose deposits of the upper part of the outcrop at site 5 (Figures 9 and 10b,d).

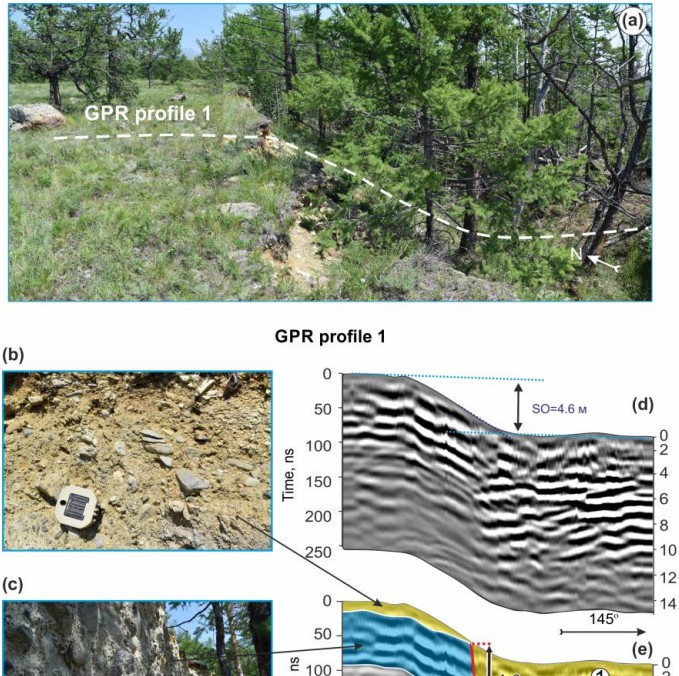

**Figure 10.** GPR profile 1 across the fault scarp (rupture R-9, see location in Figures 4 and 5): (**a**) location of GPR line; (**b,c**) reddish soft-consolidated and (**b**) well-consolidated grey predominantly coarse-grained (**c**) deposits; (**d**) radargram; (**e**) its interpretation with inferred ruptures, fault displacements (in meters), and radar facies marked by numbers in circles corresponding to layers of different dielectric properties.

The second radar facies has relatively extended reflection features and high reflection amplitudes. In the geological section, these correspond to well-consolidated gray, predom-

inantly coarse clastic deposits, which are exposed in a subvertical wall over a distance of ~300 m SW of site 5 (Figure 10c). In the place of the GPR profile, these deposits were covered with scree, which strongly leveled the scarp. This served as a favorable factor for the GPR study across the strike of the rupture. The third radar facies is characterized by weak signal amplitudes and is not exposed anywhere on the surface.

The displacement of the radar facies clearly maps the main steeply dipping rupture, along which a single-event vertical throw of 4.6 m occurred (Figure 10e). It correlates with the displacement of the original surfaces (Figure 10d). In the hanging wall, according to the separation of the reflection events, three more secondary discontinuities are inferred.

### 4.3.2. GPR Profile 2

The second GPR profile begins at coordinates 53.84817° N, 108.062° E, crosses a grassy morphotectonic scarp that undercuts the debris flow fan on the left bank of the Rita River (Figure 11), and ends at coordinates 53.84739° N, 108.06245° E. The main rupture is distinguished by the displacement of radar facies 6 and 11, a well-delineated oblique reflection event, and the presence of colluvial wedges (radar facies 3 and 5) that stand out in the GPR image. A set of secondary fractures, which do not affect the scarp height, is mapped in the hanging-wall block. Judging by the irregular features of the bottom of radar facies 1, these cracks were formed as secondary macroseismic effects from a later seismic event of a small magnitude when the fault scarp was already formed. These shallow fractures limit one of the Rita River channels that was active in 1908 [45] and was outlined on the radargram.

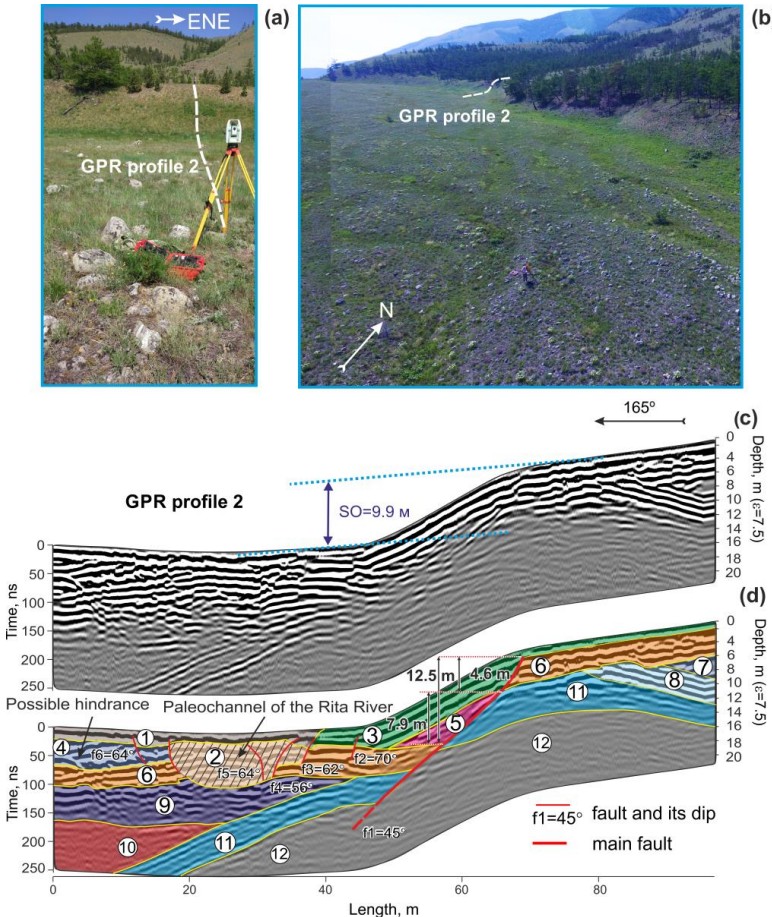

**Figure 11.** GPR profile 2 across the fault scarp (rupture 10, see location in Figures 4 and 5) cutting debris flow fan: (**a**,**b**) location of GPR line; (**c**) radargram; (**d**) its interpretation with inferred ruptures, vertical component of fault displacements (in meters), and radar facies marked by numbers in circles corresponding to layers of different dielectric properties.

In contrast to the GPR profile 1, the observed cumulative displacement of 9.9 m is significantly less than the 12.5 m throw inferred for GPR profile 2. This is explained by the burial of the scarp, noticeable on the radargram, as a result of which the surface in the downthrown block in the delta area was leveled off. Based on the relationship of the radar facies and the presence of two colluvial wedges, it follows that the throw value of 12.5 m is a sum of at least two seismic events. At an earlier paleoearthquake, a 7.9 m displacement occurred in this place; the later one was 4.6 m. It is seen from colluvial wedges interpreted on the radargram (Figure 11b).

### 4.3.3. GPR Profile 3

The GPR profile 3 begins at coordinates 53.8483° N, 108.06757° E, crosses a paleorupture at the base of seismogenic landslide II from the left bank of the Rita River, and ends at coordinates 53.84879° N, 108.06725° E (Figures 5 and 12). Due to the steep slope, the scarp quickly denudated here, and the delta plain flattened out. Consequently, stratigraphic markers correlated on both sides of the scarp are absent on this site. Nevertheless, the slope angle near the base of the slope increased, which indicates that the bottom of the landslide was indeed affected by faulting (Figure 12a,b). These observations are confirmed by the GPR image, which clearly shows oblique, near parallel reflection events associated with a 22 m wide rupture zone, where fracture planes dip from 59° to 70° to the southeast (Figure 12c,d). A particular river channel, observed in the GPR profile 2 (Figure 11) and on the map of 1908 [53,55], coincides with the marginal rupture (Figure 12). Determining the throw of that fault from the GPR profile 3 is difficult since it was apparently accumulated of small offsets on each fault.

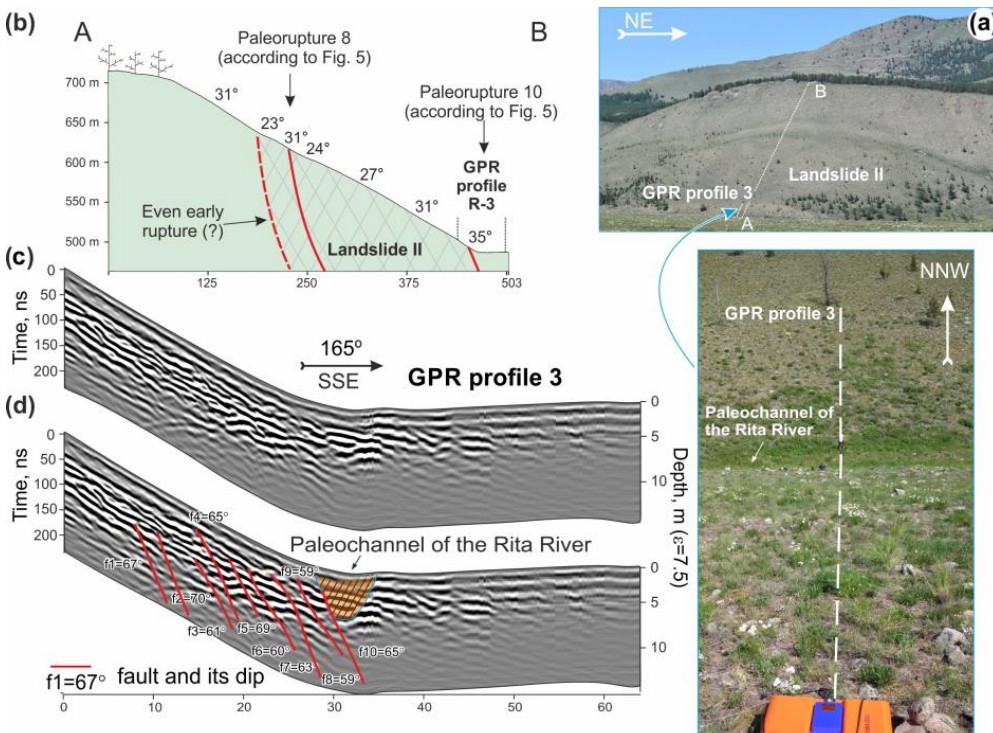

**Figure 12.** GPR profile 3 across the fault scarp (rupture 10, see location in Figures 4 and 5): (**a**) location of GPR line at the base of landslide II; (**b**) topographic profile A–B across the landslide II; (**c**) radargram; (**d**) its interpretation with inferred ruptures and paleochannel.

### 4.4. Measurements of Topographic Profiles

A very high-resolution DSM (Figure 4) was used to determine the lateral variation of displacements along the fault strike. The offset estimations were made by measuring the vertical separation between the upper and lower original surfaces on the two sides of the morphotectonic scarp located in the distal part of the Rita River delta (Figure 13d–f).

The topographic profiles (TP) are perpendicular to the fault scarp traced from 28 to 235 m from each other. The research intervals were determined by the possibility of measuring at given points since the technique is based on the analysis of parallel or near-parallel original surfaces [12]. If the slope of the two lines differed by 2° or more, the profile was discarded since it was deemed impossible to match the upper and lower surfaces in such cases adequately. The optical imagery (Figure 3) also helped to identify unmodified surfaces and to avoid vegetation masking. Consequently, vertical offsets of original surfaces, scarp slope angles, and heights were measured at 27 topographic profiles (Table 1).

**Table 1.** Parameters of paleoseismic scarps in the distal part of the Rita River delta.

| Profile Number | Distance between Profiles, m | Total Distance, m | Vertical Surface Offset, m | Scarp Slope Angle, ° | Scarp Height, m |
|---|---|---|---|---|---|
| | | Right bank of the Rita River | | | |
| 0 | 200 | 200 | - | 13 | - |
| 1 | 675 | 675 | 0.5 | 13 | 0.5 |
| 2 | 235 | 910 | 0.5 | 22 | 0.6 |
| 3 | 115 | 1025 | 0.7 | 10 | 0.7 |
| 4 | 162 | 1187 | 0.9 | 18 | 1 |
| 5 | 114 | 1301 | 0.5 | 15 | 0.6 |
| 6 | 45 | 1346 | 1.3 | 21 | 1.3 |
| 7 | 42 | 1388 | 2.3 | 24 | 2.3 |
| 8 | 47 | 1435 | 2.3 | 37 | 2.7 |
| 9 | 39 | 1474 | 1.7 | 25 | 1.7 |
| 10 | 42 | 1516 | 1 | 20 | 1.3 |
| 11 | 142 | 1658 | 1.5 | 28 | 1.5 |
| 12 | 33 | 1691 | 0.9 | 15 | 0.9 |
| 13 | 72 | 1763 | 2.8 | 36 | 2.8 |
| 14 | 45 | 1808 | 3.7 | 34 | 3.7 |
| 15 [1] | 8 | 1816 | 4.6 | 33 | 4.6 |
| 16 | 66 | 1874 | 4.9 | 70 | 5.3 |
| 17 | 36 | 1910 | 5 | 39 | 5.9 |
| 18 | 50 | 1960 | 3.4 | 78 | 3.5 |
| 19 | 50 | 2010 | 2.8 | 62 | 2.8 |
| 20 | 28 | 2038 | 3 | 56 | 3.2 |
| 21 | 45 | 2083 | 5 | 50 | 5.6 |
| | Minimum | | 0.5 | 10 | 0.5 |
| | Maximum | | 5 | 78 | |
| | Average | | 2.2 | | |
| | | Left bank of the Rita River | | | |
| 22 | 945 | 3028 | 6.5 | 25 | 7.6 |
| 23 | 47 | 3075 | 8 | 30 | 9.5 |
| 24 | 97 | 3172 | 8.2 | 30 | 11 |
| 25 [2] | 28 | 3200 | 9.9 | 33 | 12.5 |
| 26 | 52 | 3252 | 9 | 28 | 10.5 |
| 27 | 80 | 3280 | 8.5 | 34 | 11.1 |
| | Minimum | | 6.5 | 25 | 7.6 |
| | Maximum | | 9.9 | 34 | 12.5 |
| | Average | | 8.35 | | |

[1] Topographic profile 15 corresponds to GPR profile 1, where an offset of 4.6 m is a single-event offset. [2] The topographic profile 25 corresponds to GPR profile 2, where a 12.5 m vertical offset is the result of two earthquakes with displacements 7.9 m (earlier event) and 4.6 m (later event), judging from analysis of GPR facies.

Topographic measurements show an increase in the vertical surface offsets (SO) from southwest to northeast (Figure 13b). On the right bank, the maximum throw between the original surfaces is 5 m, with an average offset of 2.2 m. The GPR study on profile 1

(Figure 10), which coincides with topographic profile 15, shows that the displacements on the right bank of the Rita River are associated with a single earthquake. Displacement values obtained by two different methods are equal.

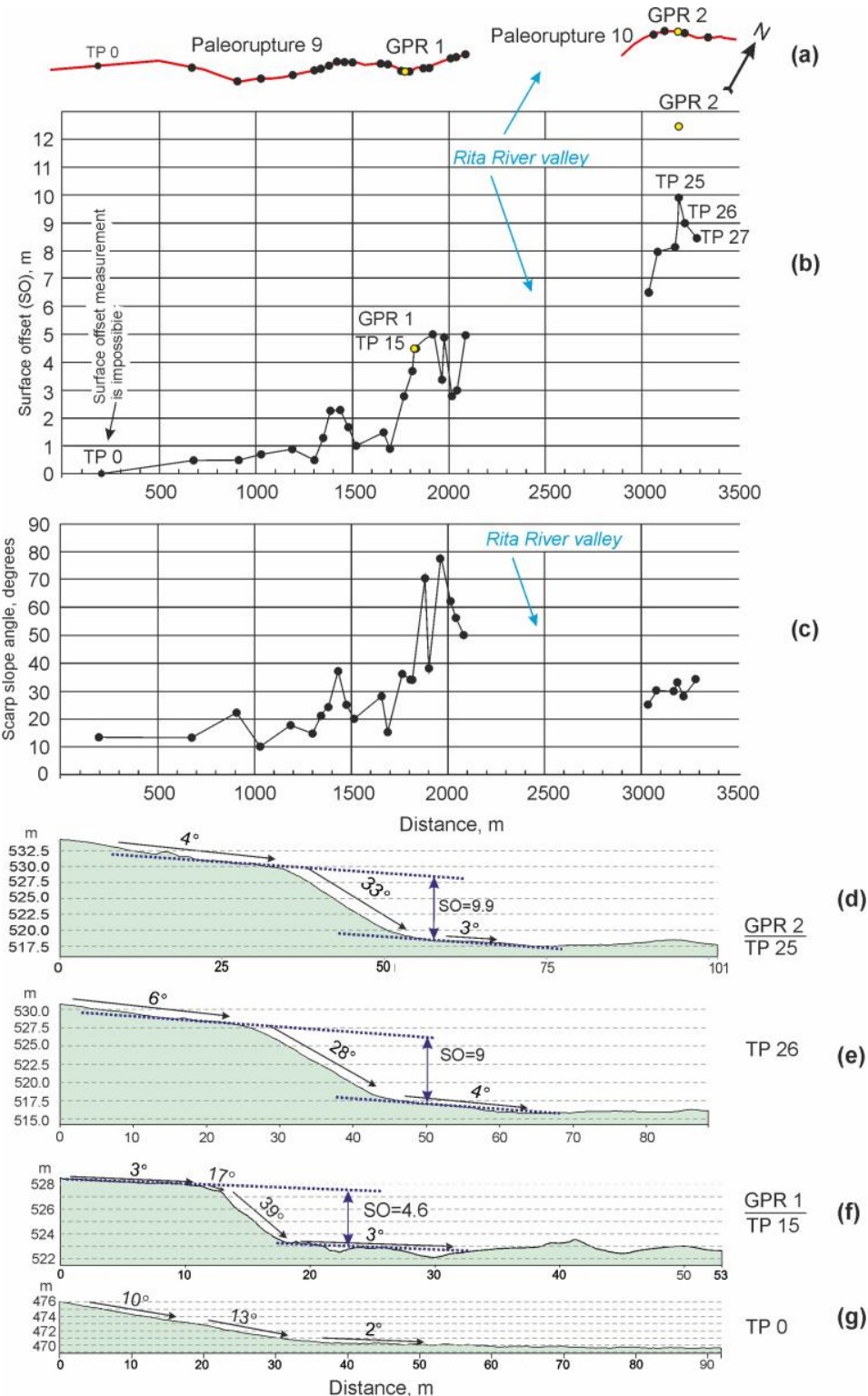

**Figure 13.** Results of topographic measurements: (**a**) paleoseismic ruptures R-9 and R-10; (**b**) changing of surface offsets along ruptures R-9 and R-10; (**c**) variation of scarp slope angle; (**d–g**) some topographic profiles (TP) across fault scarps. Black point indicates measurement of SO, yellow point shows full offset of GPR facies (see Figures 10 and 11).

To the northeast of the Rita River, the displacements of the original surfaces are significantly larger. A maximum value of 9.9 m was measured at the location of GPR profile 2 (Figure 13b,c), where, as noted above, the total vertical displacement of the GPR facies is 12.5 m (Figure 11). The offset difference of 2.6 m, estimated by various methods, is probably related to the high rate of sediment accumulation at the base of the fault scarp that cuts through a small debris flow fan (see Figures 3b and 4c). At 10 m from the GPR profile 2, at site TP 26 (Figure 13b,e), a throw measured by the displacement of the original surfaces is 0.9 m less. The slope angle of the seismogenic scarp is 5° flatter, which may indicate a more significant denudation degree and sediment accumulation on the TP profile 26, as well as a strong variability in the parameters of the seismogenic rupture even within a few tens of meters. Studies on the GPR profile 2, which coincides with the topographic profile 25, show that the offsets to the right of the Rita River are likely the result of two seismic events and confirm partial burial of the fault scarp (Figure 11).

To the northeast of TP 27, the rupture cuts off the lower part of the landslide slope, causing its dip at the base to become steeper by 4–7° (Figures 4c and 12a,b). Measuring the displacements of the original surfaces is not possible along this segment since there is significant denudation and partial burial of the fresh fault scarp by slope deposits. Consequently, the displaced original surfaces have significantly different dip angles. Additionally, the seismic paleorupture is displayed by a linear trough, which was used by one of the channels of the Rita River. The average inclination angle of the previously formed landslide slope is 30°, and in some places, it reaches 52°, which indicates a high rate of accumulation of destruction products of the massif at its foot.

It should be noted that the maximum slope angles of the paleoseismic scarp (Figure 13c) are in the central part of the ruptured system southwest of the Rita River valley. Here, the fragment of paleodelta plain has the most even surface on the whole of Cape Rytyi, which prevents the rapid flattening of the scarp. On the flanks of the cape, the slopes bordering the modern delta are steeper, and therefore the products of their destruction almost completely cover the traces of seismic slips.

## 5. Discussion

Based on the analysis of the relationship between landforms and ruptures, inferred from the interpretation of the orthophotomap and DSM of Cape Rytyi and its surroundings, two generations of coseismic paleoruptures and a pattern of recent seismically induced gravity failures in the peripheral part of the Rytyi River delta were initially revealed (Figure 5). The latter was studied in detail immediately after their discovery [53], so we mainly focused on primary paleoseismic structures in this work.

### 5.1. The Number of Seismic Events

As a result of structural and geological studies, we found out that the rupture R-1, assigned to the first generation (Figure 5) on the right bank of the Rita River, later underwent activation, expressed by the formation of fractures inside the clastic dike without noticeable fault displacements (Figure 7). In turn, rupture R-10 to the left of the Rita River (Figure 5), attributed to the second faulting generation, could have formed simultaneously with rupture R-1 since evidence of two distinct slip events separated by a colluvial wedge has been documented in this research based on GPR data (Figure 11). To the southwest of the Rita River valley, only a single coseismic throw was detected along the rupture R-9, which occurred during the second stage of the late Quaternary reactivation of the Kocherikovsky fault.

Both ruptures R-9 and R-10 undercut the debris flow fans, and the erosion slopes, which mask surface ruptures R-3, R-5, and R-6. Thus, at Cape Rytyi and its vicinity, the paleoseismic rupture zone is the result of at least two paleoearthquakes whose traces were inferred on the base of different methods. These conclusions are important for determining the correct maximum displacements and for estimating earthquake magnitudes.

### 5.2. The Magnitudes of the Seismic Events

To the southwest of the Rita River, the maximum vertical single-event offset established along the surface rupture is 5 m, while the average is 2.2 (Figure 13b, Table 1). The absolute convergence of measurements from GPR and topographic data at the same site (see Figures 10d,e and 13b, GPR 1 and TP 15) support the robustness of the obtained values. Steep slope angles at some measurement sites (Figures 10c and 13c) suggest a high degree of sediment cementation at the moment of the earthquake. To the northeast of the Rita River, the maximum of 12.5 m displacement, established from GPR data, is likely the sum of two slip events: 7.9 m for the first paleoearthquake and 4.6 m for the second one. The scarp does not have nickpoints (Figure 13d,e, TP 25 and TP 26) that indicate a relatively short recurrence interval, no more than a few thousand years [12], p. 216.

As far as we did not observe lateral component, the obtained vertical displacements and the occurrence of a paleoseismic rupture zone with a cumulative length of 37 km along the Kocherikovsky fault (Figure 2) were used to estimate the magnitude of the paleoearthquakes (Table 2). As a result, the Mw of the first event is 7.25 (Ms = 7.44); the Mw of the second event ranges between 6.93 and 7.11 (Ms = 6.94–7.28). On a small-scale seismotectonic map of Eastern Siberia, M = 7.0 was assigned to the active Kocherikovsky fault, without information about the magnitude type and the estimating approach [56]. Based on the morphostructural analysis of scarps, Chipizubov et al. [26] assumed that the seismogenic deformations extending from Cape Kocherikovo in the north direction (Figure 2) could have been formed during two paleoevents with M ≥ 7.6. In summary, the present investigation confirms the seismogenic potential of the Kocherikovsky fault, at least for the segment of the zone within Cape Rytyi.

**Table 2.** Earthquake magnitudes are estimated from empirical equations for normal faults.

| Equations [40] | $M_w$ | S | Equations [41,42] | $M_s$ | S |
|---|---|---|---|---|---|
| Earlier event (1), MD = 7.9 m | | | | | |
| $M_w = 6.61 + 0.71 \times \log MD$ | 7.25 | 0.34 | $M_s = 6.73 + 0.79 \log \times MD$ | 7.44 | 0.44 |
| Later event (2), MD = 5 m, AD = 2.2 m, L = 37 | | | | | |
| $M_w = 4.86 + 1.32 \times \log L$ | 6.93 | 0.34 | $M_s = 5.8 + 0.73 \times \log L$ | 6.94 | 0.54 |
| $M_w = 6.61 + 0.71 \times \log MD$ | 7.11 | 0.34 | $M_s = 6.73 + 0.79 \log \times MD$ | 7.28 | 0.44 |
| $M_w = 6.78 + 0.65 \times \log AD$ | 7.00 | 0.33 | | | |

Note. L—surface rupture length, km; MD—maximum displacement, m; AD—average displacement, m; $M_w$—moment magnitude; $M_s$—surface-wave magnitude; S—standard errors.

### 5.3. Chronological Constraints of the Seismic Events

Determining the age of the paleoseismic deformations on the northwestern coast of Lake Baikal is still a difficult problem to solve since it is necessary to use dating methods that do not disturb the soil cover. Cape Ryty is a uniquely protected area, even within the Baikal-Lena Nature Reserve. Rare species of birds and a large number of bears live there, annually counted by the reserve staff. In the absence of data on absolute dating, we tentatively attempted to define the period of formation of the geomorphological forms, taking as a starting point the wettest time period in the late Pleistocene and Holocene, which took place 12–14 thousand years ago in the Baikal region [57,58].

We assume that erosion processes and sediment accumulation intensified at that time, caused by the melting of the mountain glaciers between the western near-top part of the slope of the Baikal Range and its spur. A group of tarn lakes and a swampy area near the source of the Rita River indicates it (see Google Earth maps at the source of the Rita River). The traces of the most powerful floods in the Baikal region occurred 12,000–14,000 years BP [59], so the formation of the paleodelta—the previous fan of the Rita River composed mainly of debris flows that carried sediments of various sources—could be tentatively assigned to this period (Figure 14). A single terrace on the right bank of the valley (Figure 3e) might have formed by the end of the same period when, for some

reason, the erosion baseline abruptly decreased. The height of the terrace is estimated to be approximately 39 m above the modern valley floor due to its obstruction by slope deposits. It is close to the average ~39–43 m height of the summit zone of the ancient debris flow fan (Figure 4).

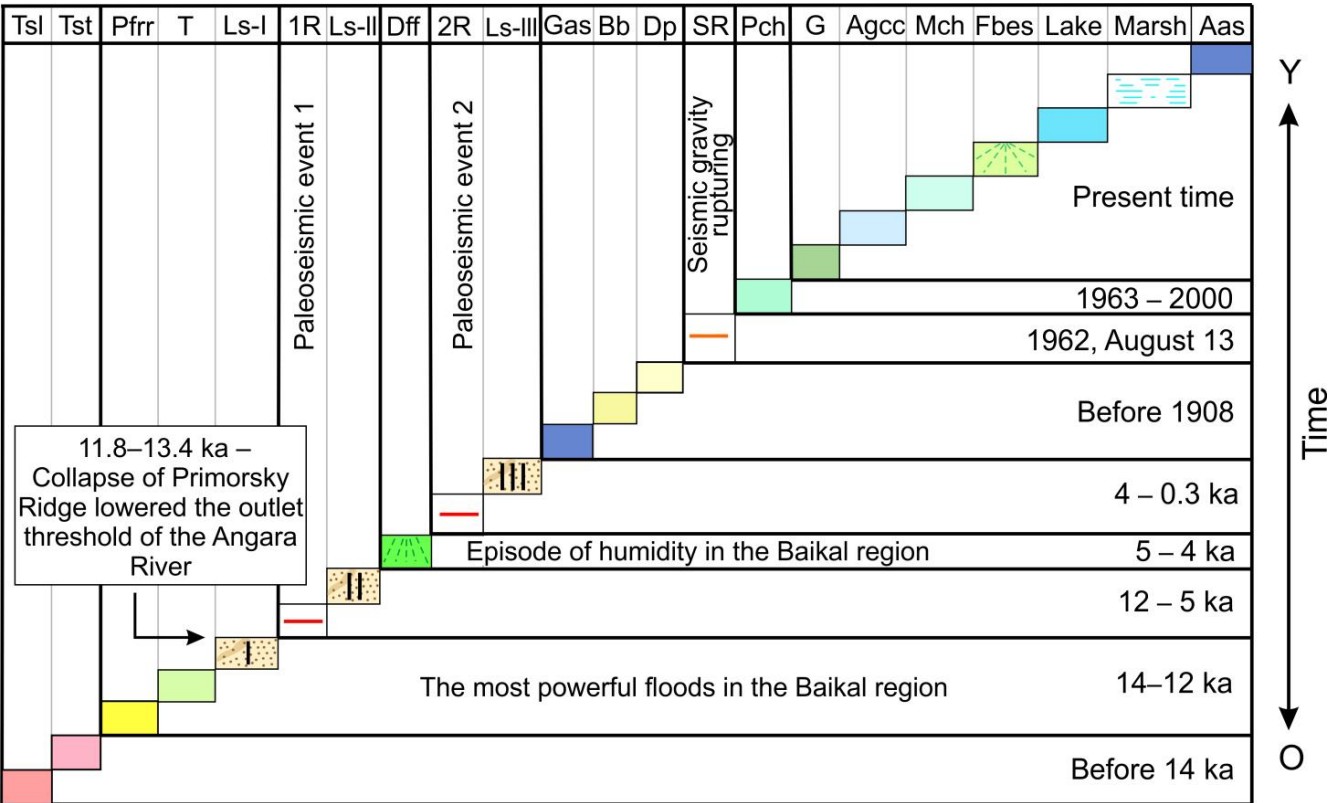

**Figure 14.** Age relationships of the mapped units in Figure 5 with the youngest at the top. Possible times of formation of units are indicated at the right. See text for further details. Tsl—tectonic slopes; Tst—tectonic step; Pfrr—Paleodelta plain (old fan of the Rita River); T—terrace of ~39 m high; Ls-I—landslide I; 1R—ruptures of first paleoseismic event activated later; Ls-II—landslide II; Dff—debris flow fans; 2R—ruptures of second paleoseismic event; Ls-III—landslide III; Gas—grass-covered abrasion scarp; Bb—braided bar; Dp—delta plain and floodplain; SR—seismic gravity ruptures; Pch—previous channels; G—gully, Agcc—accumulative gravel and cobbles coast; Mch—modern channels; Fbes—Fan and bottom of ephemeral stream; Aas—active abrasion scarp.

A sharp lowering of the erosion baseline could have occurred due to a catastrophic event in the south of Lake Baikal at the end of MIS 2 (~11.8–13.4 thousand years ago), which led to the next collapse of a part of the Primorsky Range and a lowering of the spillway threshold of the Angara River for several tens of meters [60]. Another possible reason for the dropping level of Lake Baikal could be a global climatic event at ~11.5 ka when the Marmara sea level was ~90 m down during the Younger Dryas [61].

The large landslide I identified mainly by its well-defined back wall and an approximate contour of the body was also conditionally assigned to the period of 12–14 ka since its preservation is very poor. Subsequently, against the background of a general decrease in the humidity in the Baikal region, a small surge was noted in the interval 5–4 ka BP [58,62]. Then, small debris flows, overlapping the first generation, weakly expressed ruptures R-3, R-5, R-6, R-8, and R-13 could form to the left and to the right of the Rita River valley. This means that the first rupturing earthquake in the Holocene occurred between 12 and 5 ka. The second rupturing event occurred after the formation of the mentioned debris flows since the ruptures R-9 and R-10 cut off their marginal parts, but before 1709—the time when the first Siberian records of earthquakes appeared—indicating the absence of catastrophic

earthquakes near Cape Rytyi for the last ~0.3 ka. Considering the scarp without nickpoints and the structure of the section on GPR profile 2 (Figure 11), it can be assumed that the interval between two paleoearthquakes was no more than 3 ka.

After these specified paleoseismic events, the delta deposits repeatedly underwent secondary coseismic effects from moderate earthquakes in Lake Baikal. The shallow fractures affecting the tops of the section (Figure 11) and a zone of recent seismic gravity ruptures found in the peripheral delta part (Figure 3c,d) and associated with the 1962 earthquake [53] support this interpretation.

It is preferable to use absolute methods for dating sediments and deformational events to reconstruct the complete spatiotemporal sequence of climatic and tectonic events on Lake Baikal. Such studies would be of great importance for understanding the relationship between these two processes and their effect on the development of large inland water bodies and adjacent territories.

## 6. Conclusions

Comprehensive geomorphological and structural studies of Cape Rytyi, about which numerous legends have been created, made it possible to draw the following main conclusions:

1.  Cape Rytyi and the adjacent territory are characterized by a wide variety of natural geomorphological and structural features, which indicate the widespread past development of debris flows, landslides, and rupturing processes associated with at least two paleoseismic events. Currently, in the peripheral part of the Rita River delta, a zone of seismically induced gravity ruptures formed during the remote 1962 earthquake has been observed.

2.  The surface faulting in the distal part of the Rita River delta and on the slopes of the Baikal Ridge are included in the seismic rupture pattern, which was mapped over a distance of 37 km and was associated with the activation of the Kocherikovsky fault. The width of the primary deformation zone is ~2 km. On the hanging wall side, the boundary of the shear zone can be conditionally limited by the mentioned modern seismically induced gravity failure zone at the edge of the Rita River delta, whose northeastern segment coincides along strike with the direction of paleoseismic ruptures and is located 1.5 km from them. Accordingly, the damage zone of the Kocherikovsky fault could be wide up to at least 3.5 km.

3.  The maximum displacement during the first paleoearthquake, which occurred at a 12–5 ka interval, reached 7.9 m, and 5 m during the second one, which was 4–0.3 ka ago. The magnitude estimated from different relationships using the rupture length and displacements are $M_w = 7.3$ ($M_s = 7.4$) and $M_w = 6.9$–$7.1$ ($M_s = 6.9$–$7.3$) for the first and second events, respectively.

4.  The preservation of coseismic scarps, their dip angles, and the degree of burial strongly depend on the initial landscape and can vary even within a few hundred meters. This fact must be taken into consideration when conducting morphotectonic studies to determine the rupture parameters.

High-resolution UAV survey provides new possibilities for detailed geomorphological and structural investigations. Interpretation of orthophotomaps and DSMs with a spatial resolution of a few cm/pixel enables not only mapping with accuracy but also observing the relationship between the Earth's surface features in order to estimate their relative ages. Concurrently, more accurate conclusions about the seismic history of the late Quaternary faults would require ground studies. An effective complex, alternative or complementary to trenching, can be the GPR method and geological and structural study of natural rock outcrops, which could contribute to distinguishing the number of deformation events on a specific fault.

Such information will be practical if absolute dating data are not available. In the presented article, we tried to show the advantages of combining very high-resolution aerial photography, GPR, geomorphological and structural studies, a complex of which can be applied to improving seismic hazard assessment analyses in other regions of the world.

**Author Contributions:** Conceptualization, O.V.L.; methodology, O.V.L.; software, O.V.L., A.A.G., I.A.D. and C.B.; validation, O.V.L. and A.A.G.; formal analysis, O.V.L. and I.A.D.; investigation, O.V.L., I.A.D., A.A.G. and C.B.; resources, O.V.L., A.A.G. and I.A.D.; data curation, O.V.L. and A.A.G.; writing—original draft preparation O.V.L., I.A.D., A.A.G. and C.B.; writing—review and editing, O.V.L.; visualization, O.V.L., I.A.D. and C.B.; supervision, O.V.L.; project administration, O.V.L.; funding acquisition, O.V.L. All authors have read and agreed to the published version of the manuscript.

**Funding:** The study was funded by the Russian Science Foundation grant 22-27-00064, https://rscf. ru/project/22-27-00064/ accessed on 15 March 2023.

**Data Availability Statement:** Not applicable.

**Acknowledgments:** We thank Andrei Gladkov for his help during the fieldwork and express our gratitude to Yossi Mart for his constructive comments. Special thanks go to the reviewers who contributed greatly to the improvement of the manuscript, including English editing.

**Conflicts of Interest:** The authors declare no conflict of interest.

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
