# Peer review of "Enigmatic Surface Ruptures at Cape Rytyi and Surroundings, Baikal Rift, Siberia: Seismic Hazard Implication"

_quaternary, doi:10.3390/quat6010022_

Round 1

Reviewer 1 Report

see comments 'pencilled' in the attached file

Author Response

Dear reviewer,

Thank you very much for the great work under our article. I appreciate very much  your comments, which significantly improve the manuscript. Please find responses on each of your points in attached file.

Sincerely,

Oksana Lunina

Reviewer 2 Report

The manuscript is very interesting and well written. I have few comments which are also detailed in the attached annotated manuscript.

1. please mention in the introduction the largest historical and instrumental events recorded in the Lake Baikal region and their earthquake magnitude; this will help the reader to better understand the seismotectonic framework of the area

2. it would be very useful to add a morphobathymetric map of the lake offshore in front of the Rita River delta 

3. Figure 8 : is this a gravity-graben like the ones described in Nevada by, for instance, Slemmons 1954 (Slemmons, D. B. (1957). Geological effects of the Dixie valley-Fairview peak, Nevada, earthquakes of December 16, 1954. Bulletin of the Seismological Society of America, 47(4), 353-375)? I attach here the paper for your reference; if this the case, you may want to mention this and make reference to the Slemmons paper; coseismic gravity-graben are very distinctive evidence for M7 Holocene earthquake surface faulting

4. i have edited some english language even if i am not an english mother tongue, hope this is ok for improve the final version

Author Response

Dear Reviewer,

Thank you very much for the work under our article, spent time, useful comments, and recommendations, which certainly improved the article. Please see attached file with the responses on your comments.

Sincerely,

Oksana Lunina

Reviewer 3 Report

The article is descriptive in nature, which describes the video material obtained using very high-resolution aerial photography. It is clear that for some areas of science, these research methods have a certain valuable value. But it is not clear how such studies can be used to solve the problems of seismic hazard assessment in other regions of the world? The statement is absolutely illiterate and unnecessary in this article. The authors can leave the entire research part, but should drop the mention of the possibility of using their methods to study seismic safety. There is a long-term (25-35 years), medium-term (months) and short-term (hours) earthquake forecast. What will the results presented in the article give for this? Nothing! The results in the article are not suitable for this. 

Author Response

Dear reviewer,

Thank you very much for the time you spent for our paper and comments, which improve the manuscript. I appreciate it very much. Please find response in the attached file.

Sincerely,

Oksana V. Lunina.

Reviewer 4 Report

1.       Add the scientific issues in the abstract, to let the readers clear of the purpose.  

2.       Introduction: Add international cases.

3.       Add the description to make the Ms of global interest.

4.       Add the rock association or to say the geological setting to support the following.

5.       the ages of the seismic events can be descried in detail and with clear sequences.

6.       The discussion is a bit short compared with the long results.

7.       Condense the conclusions.  

Author Response

Dear Reviewer,

Thank you very much for your work, time, and useful recommendations, which certainly improved our manuscript. Please find my responses on each of your comments in the attached file. 

Sincerely,

Oksana V. Lunina.
